

# Automated wideline nuclear quadrupole resonance of mixed-cation lead halide perovskites

Jop W. Wolffs[1], Jennifer S. Gómez[1], Gerrit E. Janssen[1], Gilles A. de Wijs[1], and Arno P. M. Kentgens[1]

[1]Radboud Universiteit, Institute for Molecules and Materials, Heyendaalseweg 135, NL-6525 AJ Nijmegen, The Netherlands

**Correspondence:** Arno P. M. Kentgens (a.kentgens@nmr.ru.nl)

**Abstract.** Nuclear quadrupole resonance (NQR) is a sister technique to NMR that is extremely sensitive to local crystal composition and structure. Unfortunately, in disordered materials this sensitivity also leads to very large linewidths, presenting a technical challenge and requiring a serious time investment to get a full spectrum. Here we describe our newly developed, automated NQR setup to acquire high-quality wideline spectra. Using this setup, we carried out $^{127}$I NQR on three mixed-cation lead halide perovskites (LHPs) of the form $MA_xFA_{1-x}PbI_3$ (MA = methylammonium, FA = formamidinium, $x = 0.25, 0.50, 0.75$) at various temperatures. We achieve a signal-to-noise of up to $\sim 400$ for lineshapes with a full-width-half-maximum of $\sim 2.5\,\mathrm{MHz}$ acquired with a spectral width of $20\,\mathrm{MHz}$ in the course of two to three days. The spectra, which at least partially exhibit features encoding structural information, are interpreted using a statistical model. This model finds a degree of MA–MA, FA–FA clustering ($0.2 \leq S \leq 0.35$). This proof-of-principle for both the wideline NQR setup and the statistical model widens the applicability of an underutilised avenue of non-invasive structural research.

## 1 Introduction

### 1.1 Wideline NQR

The quadrupolar interaction occurs when a nucleus with spin quantum number $I > 1/2$ is situated in a non-vanishing electric field gradient (EFG). In NMR, the resonance frequency of a quadrupolar nucleus depends on the orientation of this interaction with respect to the external magnetic field, resulting in specifically shaped powder patterns. The strength of the quadrupolar interaction determines the width of these patterns, and these can be tens or even hundreds of megahertz wide. Nuclear quadrupole resonance (NQR) (Das and Hahn, 1958) probes the quadrupolar interaction directly in the absence of an external magnetic field. The stationary spin states are then the eigenstates of just the quadrupolar Hamiltonian. A detailed explanation of the associated energy level diagram can be found in e.g. Seliger (1999) and Semin (2007). Here we just mention the case of spin $I = 5/2$ and EFG asymmetry $\eta_Q = 0$, which is approximately true for all NQR studies reported in this work. This gives



NQR transition frequencies

$$\nu_n = \frac{3n}{20} C_Q \tag{1}$$

where $n = 1, 2$. In terms of the spin magnetic quantum number, these are the $\pm\frac{1}{2} \leftrightarrow \pm\frac{3}{2}$ and $\pm\frac{3}{2} \leftrightarrow \pm\frac{5}{2}$ transitions. When $\eta_Q$ is not zero, these states mix and an additional transition $\nu_3 = \nu_1 + \nu_2$ becomes allowed. In this example, as in all other NQR frequencies, there is no dependence on the orientation of the EFG and a linear dependence on $C_Q$.

In well-ordered materials this results in sharp resonances. Since the sensitivity improves with the strength of the interaction, NQR is particularly suitable where NMR becomes impractical due to excessive spectral widths. When materials are disordered, however, the many unique local environments produce many different EFGs, each with their own resonance frequencies. Due to the sensitivity of NQR to structural variations, the spread of these signals can once again be several megahertz wide (Mozur et al., 2020; Aebli et al., 2021). While this is still within reach of a proper wideline setup, and while the lack of a powder pattern makes the spectrum significantly simpler than its NMR equivalent, the technical challenges and time investment of acquiring wideline NQR spectra have so far limited its application.

## 1.2 Lead halide perovskites

Over the last few decades, NQR has re-emerged from relative obscurity, in large part due to its usefulness in studying lead halide perovskites (LHPs) (Volkov et al., 1969; Xu et al., 1991; Senocrate et al., 2018; Yamada et al., 2018; Piveteau et al., 2020a; Mozur et al., 2020; Aebli et al., 2021; Doherty et al., 2021). The exceptional properties of this class of materials have attracted attention for applications in LCD technologies, LEDs, lasers, photodetectors and more (Piveteau et al., 2020b). Perhaps the most intense interest comes from research into photovoltaics: over the last 15 years perovskites in solar cells have gone from a sensitising dye yielding an efficiency of $< 3.81\%$ (Kojima et al., 2009) to inverted solar cells with an efficiency of $> 26.1\%$ (Chen et al., 2024), establishing a role as an 'emerging' photovoltaic material (NREL, 2024).

Part of the appeal of LHPs lies in their capacity for being compositionally engineered. Perovskites are a family of crystals whose 'ideal', cubic structure is $A^{XII}_{[m\bar{3}m]} B^{VI}_{[m\bar{3}m]} X^{II+IV}_{3[4/mm]}$ (Bhalla et al., 2000), where the superscripts indicate the coordination number of each ion and the bracketed subscript the site symmetry. In lead halide perovskites, $B = Pb^{2+}$ and $X = Cl^-$, $Br^-$ or $I^-$. Common occupants of the A site are $Cs^+$ or small organic cations like formamidinium ($FA^+$) or methylammonium ($MA^+$). The latter are referred to as *hybrid* perovskites due to the mix of organic and inorganic components. All three sites can be made to contain a mix of ions to achieve certain benefits, such as a different bandgap (Noh et al., 2013; Zhao et al., 2017) or improved stability (Dai et al., 2016; Gong et al., 2018). Despite the promise of these mixed perovskites and a significant amount of previous research, there is still uncertainty regarding their structural and dynamic properties (Grüninger et al., 2021; Piveteau et al., 2020b), hindering the search for solutions to issues such as photo-induced halide segregation (Hoke et al., 2015) and instability of the perovskite structural phase (Chen et al., 2016; Yamada et al., 2018).

Due to the strong EFG between the lead ions and the large quadrupolar moments of the halides, their quadrupolar interactions are particularly strong, up to half a gigahertz for the iodides. While this makes them very suitable for NQR, which is very sensitive to local structure, its application has mostly been limited to LHPs with no or little mixing, such as $FA_x Cs_{1-x} PbI_3$





where $x \geq 0.9$ (Aebli et al., 2021). For more equal mixes, the NQR spectrum becomes too wide to easily acquire with regular, commercially available equipment.

### 1.3 Goal

Here, we seek to make wideline NQR of disordered materials easily accessible. We demonstrate and validate a newly-built, automated wideline NQR setup. We illustrate its usefulness with measurements of the second NQR resonance of $^{127}$I in mixed-

cation LHP $MA_xFA_{1-x}PbI_3$ where $x = 0.25, 0.50, 0.75$ at a temperature range of 293–420 K. By employing an automated matching and tuning robot (Pecher et al., 2017) and properly optimising the acquisition parameters, it is possible to acquire Variable Offset Cumulative Spectra (VOCS) (Massiot et al., 1995; Tong, 1996) of hundreds of subspectra per day with an overal signal-to-noise (S/N) of $\geq 100$, while requiring very little work from the operator.

The speed and ease of acquisition of wideline NQR spectra this setup achieves, enable a detailed investigation of the spectra

of strongly mixed perovskites. We show a proof-of-concept version of such an analysis, to be refined and expanded upon in future work. On the phenomenological level, we construct a preliminary model that relates cation distributions to a spectral shape, to be fitted to experimental results. On a more fundamental level of theory, we perform Density Functional Theory (DFT) calculations on structures provided by the Molecular Dynamics (MD) trajectory of $MA_{0.50}FA_{0.50}PbI_3$ from Grüninger et al. (2021), and compare observations with those of the phenomenological model.

Finally, we briefly showcase possible applications of the experimental setup to related compounds, by measuring the mixed-anion and double-mixed perovskites $MAPbI_2Br$ and $MA_{0.15}FA_{0.85}PbI_{2.55}Br_{0.45}$. However, these are not subjected to modelling in this publication.

Taken together, we hope that these experimental and theoretical findings demonstrate the viability of this laboratory setup, and its potential to contribute to the structural investigation of disordered LHPs and similar materials.

## 2 Description of the models


The new laboratory setup facilitates the acquisition of wideline NQR spectra, but these are only useful if the structural information they contain can be extracted. In order to demonstrate the potential of the setup we describe preliminary models for the interpretation of the spectra. Their purpose is to relate spectral features to nearest neighbour ion substitutions and the degree of order in the overall ion distribution.

### 2.1 The phenomenological model


The phenomenological model is constructed as a function that can be fitted to the spectra. It directly simulates an NQR spectrum based on parameters that link particular configurations of the competing $MA^+$ and $FA^+$ cations in $MA_xFA_{1-x}PbI_3$ to NQR resonance peaks. The model separates the compositional disorder following from the competing $MA^+$ and $FA^+$ cations in $MA_xFA_{1-x}PbI_3$ into a short range and a long range component. The short range component considers the NQR frequency

and probability of a particular occupation of the lower cation coordination shells surrounding a halide. Each of these particular





occupations, or 'short-range coordinations', correspond to a particular EFG. In the model, they are represented by a set of Lorentzians whose centre frequencies reflect their assumed EFGs and whose areas reflect their assumed probabilities.

For simplicity, coordinations with the same number of $MA^+$ ions but different distributions are swept together. Following binomial statistics, the unbiased 'fractional population' of a coordination in the $n$-th shell containing $k_n$ $MA^+$ cations and $(N_n - k_n)$ $FA^+$ cations is then

$$p_{k_n}^n = \binom{N_n}{k_n} x^{k_n} (1-x)^{N_n-k_n} \tag{2}$$

where $N_n$ is the total number of A cation sites in the $n$-th coordination shell. Following Tycko et al. (1992), a structural order parameter $S$ (Cullity and Stock, 2014) is introduced that describes the tendency for coordination shells to be preferentially occupied by one type of cation. Eq. (2) becomes

$$p_{k_n}^n(S) = \binom{N_n}{k_n} \left( x \, r_{\mathrm{MA}}^{k_n} (1-r_{\mathrm{MA}})^{N_n-k_n} + (1-x) \, r_{\mathrm{FA}}^{N_n-k_n} (1-r_{\mathrm{FA}})^{k_n} \right) \tag{3}$$

where

$$r_{\mathrm{MA}} = x + S(1-x) \tag{4}$$

$$r_{\mathrm{FA}} = (1-x) + Sx \tag{5}$$

$S$ varies between 0, indicating completely random distribution of the A site cations, and 1, indicating complete phase segregation into $MAPbI_3$ and $FAPbI_3$. Intermediate values indicate partial clustering of cation species. The fractional population of a specific combination of coordination shells is just the product of the fractional populations per shell $\prod_n p_{k_n}^n(S)$, where it is assumed $S$ is identical for all shells.

The NQR frequency of a particular short-range coordination is taken to be a 'base' frequency plus offsets that scale linearly with the numbers of $MA^+$ ions in the coordination shells. Mathematically, the frequencies are given by

$$\nu_2(\{k_n\}) = \nu_0 + \sum_n \Delta\nu_{\mathrm{MA}}[n] k_n \tag{6}$$

where $\Delta\nu_{\mathrm{MA}}[n]$ is the frequency shift per $MA^+$ (instead of $FA^+$) in the $n$-th coordination shell of $^{127}I$, and $k_n$ the number of $MA^+$ in this shell. Note again that these frequencies are independent of how the $k_n$ $MA^+$ are distributed in their respective coordination shells.

The number of possible coordinations increases dramatically with the number of shells. At the same time it is to be expected that shells at long range from the halide will have a small effect on the EFG. Anything beyond the first few shells is therefore not considered explicitly, but as a separate long range component of the model. A common and physically grounded approach to describe the influence on the EFG of a large number of elements at long range is the extended Czjzek distribution (Czjzek et al., 1981; Caër et al., 2010), where the EFG of the short range coordination serves as the local, fixed EFG in the extended Czjzek distribution. In the case where the local EFG is sufficiently large and sufficiently symmetric, as is the case for the halides in perovskites, the NQR spectrum of an extended Czjzek distribution simplifies to a Gaussian distribution of the resonances associated with the local EFG. This reduces the simulation time of the model significantly.





To summarise, the phenomenological model consists of $4+n_{\mathrm{max}}$ parameters, where $n_{\mathrm{max}}$ is the number of shells considered in a short range coordination:

$\nu_0$  the base frequency corresponding to zero $\mathrm{MA}^+$ ions in the short range shells.

$S$  the order parameter.

$\Gamma_{\mathrm{L}}$  the full-width-half-maximum of the Lorentzian line broadening.

$\Gamma_{\mathrm{G}}$  the full-width-half-maximum of the Gaussian expansion.

and $n_{\mathrm{max}}$ parameters $\Delta\nu_{\mathrm{MA}}[n]$ for the frequency shift per $\mathrm{MA}^+$ ion in $n$-th shell of a short range coordination.

Finally, experimental differences in intensities as a function of frequency and temperature must be taken into account. The
intensity scales with the difference in Boltzmann equilibrium population between the energy levels involved in the resonance.
The total simulated spectrum spectrum is scaled accordingly as a function of resonance frequency and temperature.

## 2.2 DFT-based models

The relation between short-range coordination and NQR frequency (Eq. (6)) is the cornerstone of the phenomenological model.
It is an assumption whose validity cannot be confirmed by the phenomenological model itself, and should therefore be verified
using a higher level of theory. DFT calculations can be employed to study this relation independently. Molecular dynamics
(MD) trajectories would provide a realistic model, but these are computationally expensive. The feasible lengths of such trajec-
tories, and therefore their statistical accuracy, is poor. We present one such trajectory, but add two simplified, but statistically
stronger models that focus on a specific interaction in the crystal lattice. All of these are based on the molecular-dynamics
(MD) trajectory described by Grüninger et al. (2021) of the simulation of a $4\times4\times4$ supercell of $\mathrm{MA}_{0.50}\mathrm{FA}_{0.50}\mathrm{PbI}_3$ with
random cation occupancy at $400\,\mathrm{K}$. In each case the EFG tensors of all 192 iodide ions in the supercells are calculated in the
'lab' frame, then diagonalised to yield the EFG principal components and NQR frequency $\nu_2$ for each individual $^{127}\mathrm{I}$ ion.

**DFT-1**  In model DFT-1 we calculate the iodide EFG tensors on the MD trajectory for all iodide ions at regular intervals. For
each $\mathrm{I}^-$ the time-average of the EFG tensor on the MD trajectory is calculated (in the 'lab' frame). This model accounts
for all motion of all ions and is therefore in principle very realistic. As mentioned, however, the trajectories are rather
short at $100\,\mathrm{ps}$, and imperfect averaging of the tensors is expected.

**DFT-2**  This simplified model probes the effect of the distortions in the inorganic backbone. To this end, Pb ions are placed
at the time-averaged positions of the MD trajectory, with the iodide ions exactly halfway between nearest-neighbour
Pb ions. The A site cations are kept at their time-averaged positions as well, but replaced with Cs ions to minimise
interactions based on cation species. This structure is not relaxed. The effect of nature and shape of the cations on the
iodide EFG tensors enters only indirectly via the positions of the Pb ions.

**DFT-3**  Complementary to DFT-2, this model aims to assess only the effect of the presence of specific organic cations. The Pb
ions are placed on an ideal cubic lattice whose lattice constant is determined by the average lattice parameters of the MD





trajectory. The $I^-$ are again fixed halfway in between. We model the cations as 'effective' FA (MA) species occupying 12 (24) symmetry equivalent positions in the $Pb_8I_{12}$ cubes: The effect of a single cation placement on the $I^-$ tensors is determined by replacing all cations as present in the MD supercell by Cs ions, except for a single MA or FA cation. The 12 $I^-$ of the cage around the organic cation are relaxed. The effect of one effective cation is obtained by averaging the EFG over the 12 (24) cation orientations with their corresponding distorted cages. In this way we obtain the contribution to all iodide EFGs as a difference to a situation with only Cs cations. We loop over all cation sites and add the effect of each effective cation to all iodide tensors.

## 3 Methods

### 3.1 Density Functional calculations

DFT calculations were carried out with the Vienna *Ab initio* Simulation Package (VASP) (Kresse and Furthmüller, 1996) using the projector augmented-wave (PAW) method (Blöchl, 1994; Kresse and Joubert, 1999) and the Perdew-Burke-Ernzerhof (PBE) exchange-correlation potential (Perdew et al., 1996, 1997). Electric Field Gradients (EFGs) were calculated using Petrilli et al. (1998). The Brillouin Zone was sampled with only the $\Gamma$-point in a $4 \times 4 \times 4$ supercell (or equivalent in smaller cells). The PAW data sets had frozen [Xe], [Kr]$4d^{10}$, [Kr]$4d^{10}$, $1s^2$ cores for Pb, I, Cs, and C respectively. The $^{127}$I quadrupole moment was taken from Pyykkö (2008). For the EFG calculations the convergence threshold was $10^{-6}$ eV for the $4 \times 4 \times 4$ supercell. Structural relaxations were carried out with a convergence threshold of $10^{-8}$ eV. The kinetic energy cutoff on the plane wave expansion was 500 eV.

### 3.2 Wideline NQR setup

NQR experiments were carried out using a home-built, single channel probe based on a chemagnetics probe housing with a horizontal solenoid coil made of 0.8 mm silver plated copper wire with an internal diameter of 5 mm. The temperature was regulated with a Chemagnetics temperature controller, calibrated with a thermocouple and confirmed by comparing the basal $^{127}$I $\nu_2$ NQR resonance frequency of MAPbI$_3$ with Yamada et al. (2018) up to 420 K. Tuning and matching was done by the H/F—X eATM robot from NMR Service GmbH (Pecher et al., 2017), adapted to the tuning and matching rods of the probe. Briefly put, the ATM robot minimises the standing wave ratio (SWR) of a low power continuous wave at the offset frequency of the subsequent acquisition. It does so by rotating the tuning and matching rods that would otherwise need to be adjusted by hand. A schematic representation of the complete variable temperature, automatically tuned NQR setup is shown in Fig. 1. Using three easily replaceable capacitors, the setup has a range of 140–182 MHz. Consistent power output across this range at high (150 W) and low (10 mW) power levels was calibrated using a Bird powermeter and the eATM robot console respectively. An in-house Python script was employed to control frequency offsets, power levels and the alternation of acquisition and matching and tuning during experiments and power calibration. Acquisition was done using a Varian VNMRS console and VnmrJ version 4.2 revision A. The external magnetic field strength at the probe location varied between 0.2 and





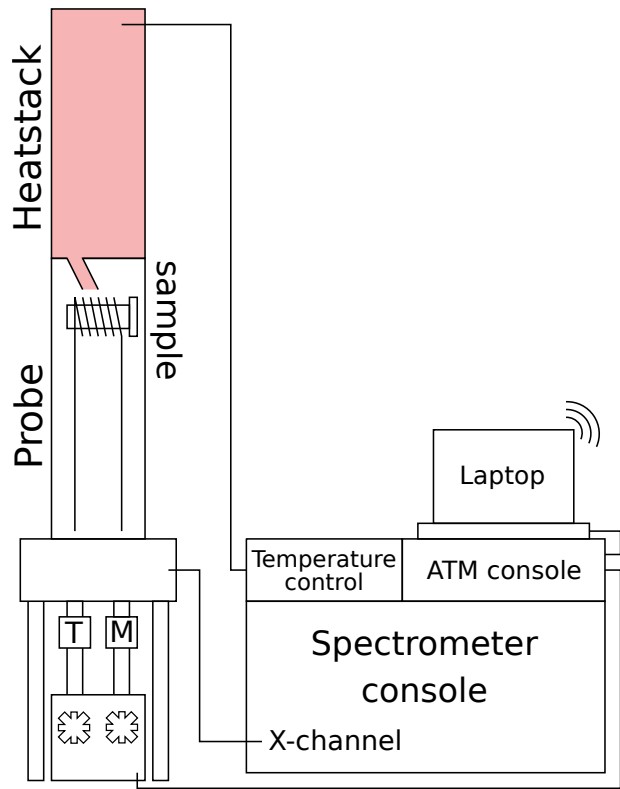

**Figure 1.** Schematic picture of the experimental setup for NQR measurements of mixed-cation samples using the Varian VNMRS console, including temperature control and automatic matching and tuning. Control of the eATM robot is available through an internet connection to the eATM-dedicated laptop.

0.8 gauss depending on orientation. Both location and orientation of the probe were kept constant throughout validation and experiments.

### 3.3 NMR and NQR measurements

Three samples were studied in detail: $MA_{0.75}FA_{0.25}PbI_3$, $MA_{0.50}FA_{0.50}PbI_3$, and $MA_{0.25}FA_{0.75}PbI_3$. In addition, we acquired initial measurements of $MAPbI_2Br$ and $MA_{0.15}FA_{0.85}PbI_{2.55}Br_{0.45}$. All were synthesised through ball-milling of mixtures of methylammonium iodide (MAI), formamidinium iodide (FAI), methylammonium bromide (MABr), lead iodide ($PbI_2$), lead bromide ($PbBr_2$) or combinations of their perovskite products. These samples are the same as in Grüninger et al. (2021), which also describes the synthesis in more detail. These samples were stored in the dark under inert atmosphere, while transfer to sample rotors was carried out in normal laboratory atmosphere and lighting. For the $MA_xFA_{1-x}PbI_3$ samples, the sample crystal structures were determined through x-ray diffraction to be the cubic $I4/mcm$ space group. After shipping, the ratios of the cations were confirmed by [1]H MAS NMR to fall within 3 percentage points of the nominal ratios (see Table S2).





**Table 1.** Acquisition parameters for all $^{127}$I NQR spectra.

| Parameter | Value |
| --- | --- |
| Temperature (K) | 293, 300–420 |
| Rotor diameter (mm) | 5 |
| Pulse sequence | Hahn echo VOCS |
| $\frac{\pi}{2}$ pulse length (μs) | 1.8 |
| $\pi$ pulse length(μs) | 3.5 |
| Echo delay (μs) | 5 |
| Recycle delay (ms) | 6[a] |
| Spectral width (kHz) | 2500 |
| Excitation width (kHz) | 225 |
| VOCS step size (kHz) | 50–200 |
| Number of points | 5000 |
| Number of scans | $2^{14}$–$2^{16}$ [b] |

[a] 16 ms for $MA_{0.75}FA_{0.25}PbI_3$ at 300 K. [b] $2^{16}$ for room temperature measurements, $2^{14}$ for 300–420 K.

190   NMR experiments were performed using a Magnex 850 MHz magnet ($B_0$ = 19.97 T) equipped with a Bruker Avance NEO console and a Varian 3.2 mm HXY MAS probe in double resonance mode. $^1$H spectra were recorded using a one-pulse experiment where rf field strength calibration and chemical shift referencing for all samples was done on powdered adamantane ($\delta_{iso}(^1H)$ = 1.756, 1.873 ppm). The spectra and acquisition parameters can be found in Supplementary section A. The samples were packed in Revolution NMR zirconia rotors.

195   For the NQR experiments the samples were packed in quartz tubes with Teflon spacers and caps. All NQR spectra are recorded as VOCS(Massiot et al., 1995; Tong, 1996) consisting of Hahn-echo ($\frac{\pi}{2}$—$\tau$—$\pi$) spectra where both $\frac{\pi}{2}$- and $\pi$-pulse length were optimised for maximum signal strength on the 164.093 MHz $^{127}$I NQR resonance of $MAPbI_3$. Using the same resonance, the full-width-half-maximum (FWHM) of the excitation profile of the Hahn-echo was determined to be 225 kHz. More acquisition parameters can be found in Table 1. The range of detected frequencies was between 159 and 179 MHz for all

200   $MA_xFA_{1-x}PbI_3$ samples and between 159 and 184 MHz but deviated for $MAPbI_2Br$ and $MA_{0.15}FA_{0.85}PbI_{2.55}Br_{0.45}$ (140–175 and 159–184 MHz respectively). In all cases the entire range of frequencies of the second NQR resonance of $^{127}$I ($\nu_2$) of the sample was covered. All processing was done in ssNake version 1.4 (van Meerten et al., 2019). Processing involved automated Lorentzian apodisation, zeroth-order phasing and conversion of data points to a common frequency axis.





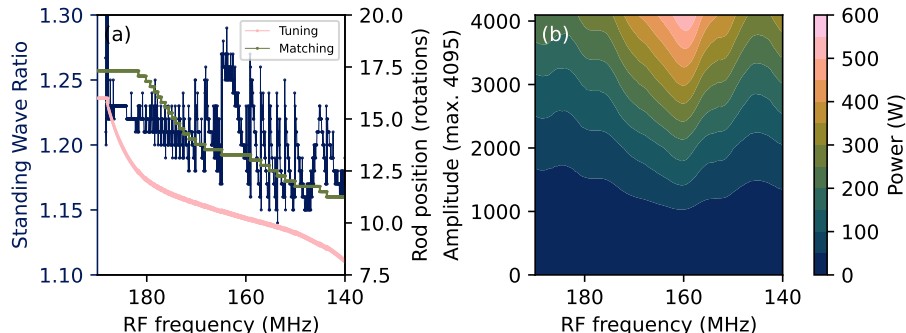

**Figure 2.** (a) Standing wave ratio and tuning and matching rod positions recorded during acquisition of the VOCS of an empty coil. (b) The power as a function of the RF frequency and the amplitude parameter in VnmrJ, recorded at of 25%, 50%, 75% and 100% of the maximum amplitude. The underlying data are available at Wolffs et al. (2025).

## 4    Results and discussion

### 4.1    Wideline NQR calibration

To show the validity of using the automated wideline NQR setup for quantitative measurements, the effective power as a function of offset frequency is characterised. Quantitative measurements require as little variation in the effective power as possible across the frequency range studied. The effective power can be affected by power reflected due to imperfect matching and tuning, the extent of which can differ as a function of offset frequency and depends on the probe. It is also not guaranteed that the power output of the spectrometer is constant over a large frequency range, but this can be compensated for by adjusting the power settings accordingly.

As part of every NQR VOCS acquisition, the reflected power is recorded in the form of the standing wave ratio by the eATM robot before the acquisition of each subspectrum. An example is shown in Fig. 2a. The standing wave ratios of 1.15–1.30 correspond to a power reflection of 0.5–1.7% across the frequency range. The upper frequency limit of the current configuration is determined by the edge of the matching capacitors at $\sim 17.5$ rotations, and was found to be $\sim 182.5\,\mathrm{MHz}$. Below $\sim 140\,\mathrm{MHz}$ the dip in the reflectance curves rapidly disappears entirely.

The power output of the spectrometer as a function of offset frequency is measured whenever any component of the setup changes. Figure 2b shows the relation between the VnmrJ amplitude parameter and the measured power output. Compensation for this inconsistency is integrated into the procedure for all measurements using the NQR setup.

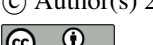

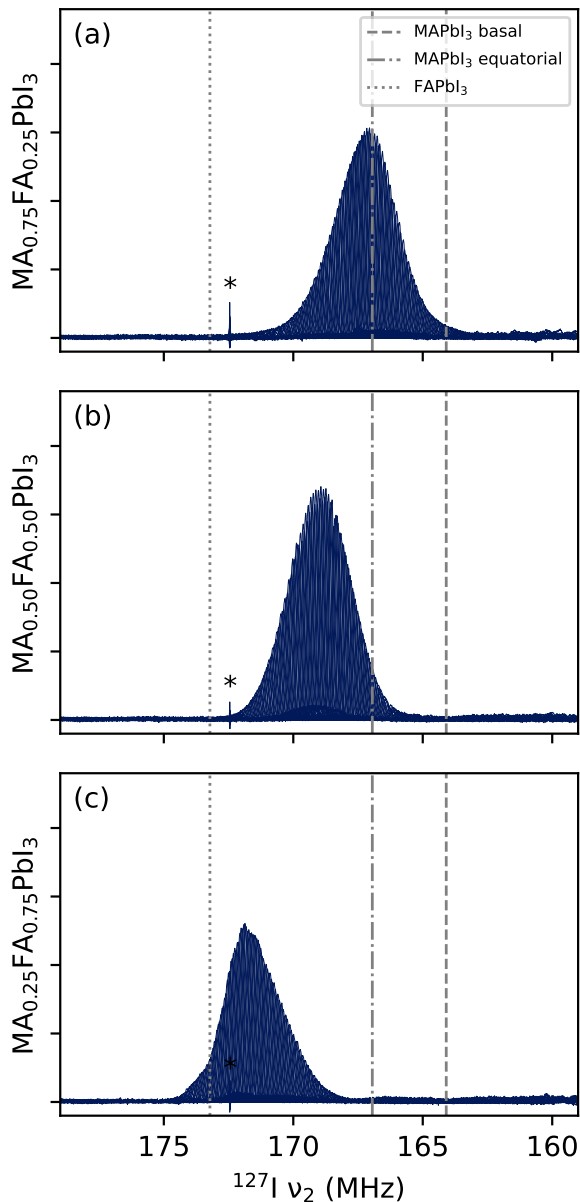

**Figure 3.** NQR VOCS at room temperature of (a) $MA_{0.75}FA_{0.25}PbI_3$, (b) $MA_{0.50}FA_{0.60}PbI_3$ and (c) $MA_{0.25}FA_{0.75}PbI_3$. Dashed and dash-dotted lines indicate the two resonances of the tetragonal $MAPbI_3$. Dotted lines indicate the resonance of the cubic $FAPbI_3$ as taken from Yamada et al. (2018) All spectra show a sharp feature around $172.5\,\mathrm{MHz}$, indicated by an asterisk, presumed to be some sort of external radio signal. Intensities should not be compared between spectra due to small differences in experimental conditions. The roughness of the skyline in (b) is due to small acquisition imperfections for a subset of subspectra, not observable in (a) and (c) and easily compensated for during processing. The underlying data are available at Wolffs et al. (2025).





## 4.2 $^{127}$I NQR at room temperature

The complete set of room temperature VOCS spectra for the three LHP samples are shown in Fig. 3. Four hundred subspectra could be recorded over the course of a weekend with, for $MA_{0.50}FA_{0.50}Pb_3$ an overall S/N ratio of $\sim 400$. Attention required from the operator was limited to a little surveillance. By comparison, earlier attempts to record these spectra manually and with less optimisation required around half an hour per subspectrum and involvement from the operator at every frequency change.

As can be expected based on the resonance of the unmixed compounds (indicated in the figure), the resonances shift to higher frequencies with increasing formamidinium content. Note that this is accompanied by an increase in the lattice constant (Weber et al., 2016). At fixed composition such a lattice expansion is expected to decrease the resonance frequency, because the slope of the electric field decreases with distance between the charges. Here, however, some property of the formamidinium evidently counteracts and outweighs this effect. In addition, the spectra appear 'skewed' towards the pure resonance of its majority cation, with $MA_{0.50}FA_{0.50}PbI_3$ being completely symmetrical. This a common phenomenon in binomial distributions and therefore not surprising.

Less expected, and therefore more interesting, is the difference in shape between the peaks of $MA_{0.75}FA_{0.25}PbI_3$ and $MA_{0.25}FA_{0.75}PbI_3$. While the former is fairly smooth, the latter exhibits (reproducible) features, particularly at the top of the peak. These features, and the fact that they are not mirrored in a compound with opposite cation ratios, show that even extremely broad NQR spectra are a source of information concerning the disorder in the lattice, if properly interpreted.

## 4.3 $^{127}$I NQR at increased temperatures

Fig. 4 shows the spectra of the three perovskite samples at different temperatures. In these and following figures, only the point of highest intensity per VOCS subspectrum is shown. For all spectra, an increase in temperature corresponds to a shift to lower frequency and a decrease in intensity. The former is consistent with thermal expansion increasing the distance between ions involved in the EFG. The decrease in intensity is mostly caused by the smaller spin state population differences at thermal equilibrium at higher temperatures and lower frequencies. To compensate for this effect, the intensity of each data point can be divided by the population difference at thermal equilibrium of two energy levels whose energy separation corresponds to the frequency of the data point. The integrated intensity after this operation are shown in Fig. 5a. There remains no clear relation between temperature and signal strength, or peak FWHM (Fig. 5b) for $MA_{0.25}FA_{0.75}PbI_3$ or $MA_{0.50}FA_{0.50}PbI_3$. For $MA_{0.75}FA_{0.25}PbI_3$, the intensity decreases and the FWHM increases, pointing towards increased relaxation.

It is interesting to see how the shapes of the peaks in Fig. 4 change with temperature. The spectrum of $MA_{0.75}FA_{0.25}PbI_3$ becomes less symmetric (it increasingly skews to low frequencies), whereas that of $MA_{0.25}FA_{0.75}PbI_3$ becomes more symmetric, and that of $MA_{0.50}FA_{0.50}PbI_3$ stays symmetric. These shape changes indicate a temperature dependent property in the material that itself depends on the cation composition, which could for example be a temperature dependent degree of local order. It is also noteworthy that the spectral features of $MA_{0.25}FA_{0.75}PbI_3$ become more pronounced at higher temperatures. While this probably plays a role in the irregular trend in Fig. 5b, it also points to some temperature-dependent property. These

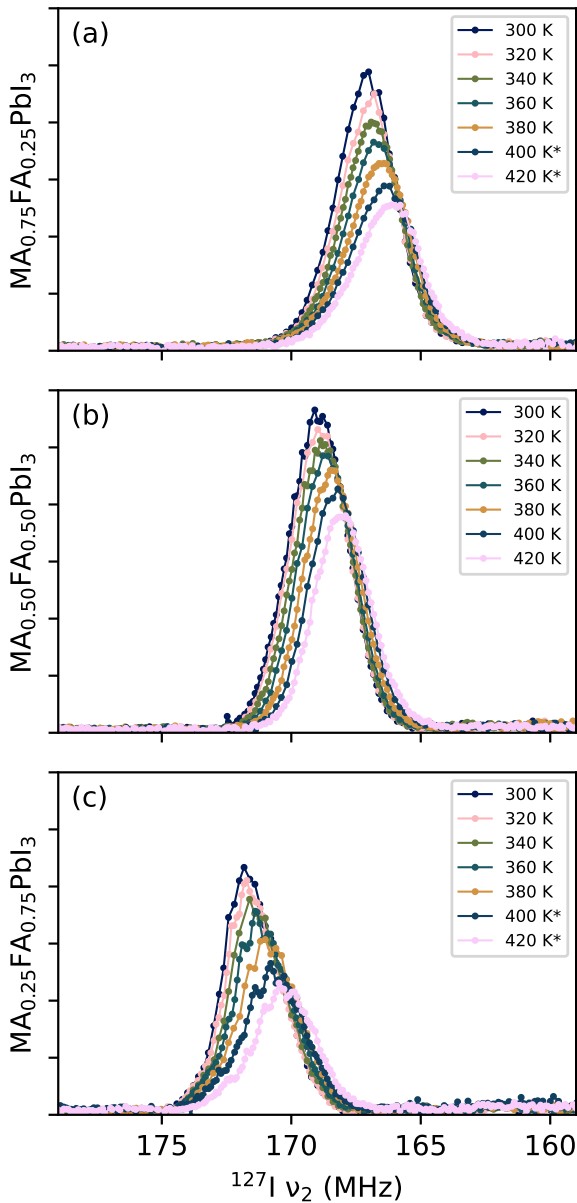

**Figure 4.** Variable temperature NQR VOCS for (a) $MA_{0.75}FA_{0.25}PbI_3$, (b) $MA_{0.50}FA_{0.50}PbI_3$ and (c) $MA_{0.25}FA_{0.75}PbI_3$. Starred spectra are more detailed repeat measurements. Their intensities, have been scaled to match the original measurements at that temperature. The underlying data are available at Wolffs et al. (2025).

observations, together with those of Sec. 4.2, illustrate the information present in NQR spectra of these kind of samples. The extraction of information, however, requires novel models.





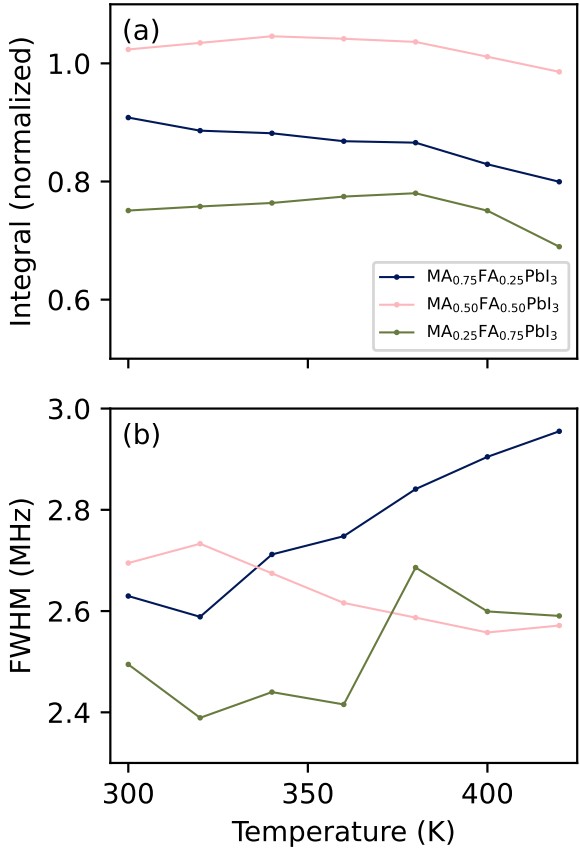

**Figure 5.** Properties of the spectra in Fig. 4 after scaling intensities to compensate for temperature and frequency. (a) Integrals over the scaled intensities, normalised to the total integral of a identically acquired VOCS of MAPbI$_3$ at 320 K (see Fig. S2). (b) Full-Width-Half-Maximum of the scaled spectra. The underlying data are available at Wolffs et al. (2025).

## 4.4 Wideline NQR modelling

### 4.4.1 The phenomenological model

The great compositional disorder of these cation mixes drives the spectral shape towards featureless humps. This particularly holds for MA$_{0.50}$FA$_{0.50}$PbI$_3$ which, combined with the symmetry of its cation mix, looks very much like a single Gaussian. There is no point in applying the model proposed in Sec. 2.1 here, because there will be many solutions of similar quality but wildly different parameter values. Fortunately, fits of the spectral shape of the other two samples, especially MA$_{0.25}$FA$_{0.75}$PbI$_3$, are particular enough to the model parameters that some observations can be made.

The first and strongest observation concerns the relative influence of short range coordination shells of A site cations. At 420 K, the spectrum of MA$_{0.25}$FA$_{0.75}$PbI$_3$ shows more than five peaks. Since the first shell around an iodide consists of four





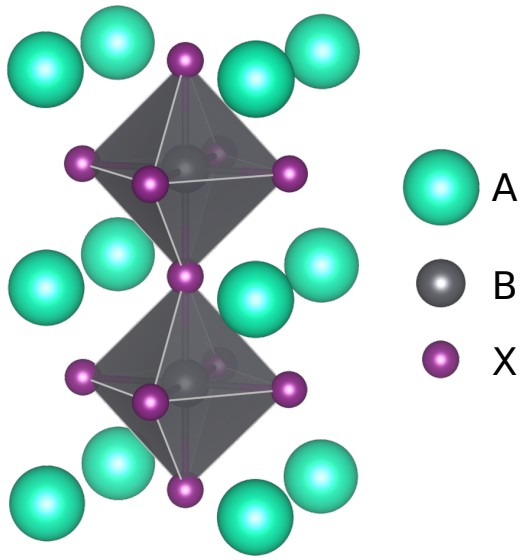

**Figure 6.** The cubic perovskite crystal structure including the first and second shells of A site cations with respect to the X site anion. Image made with VESTA (Momma and Izumi, 2011).

A site cations (see Fig. 6), this is more than the five possible MA:FA ratios of this shell. Clearly, at least one more coordination shell has to have a distinctive influence on the resonance frequencies. The phenomenological model (2.1) therefore needs to
include two shells in its short range coordinations. With the addition of the eight A site cations of the second shell, the model consists of 45 peaks.

The second observation concerns the sign of the frequency shift per MA in the first ($\Delta\nu_{\mathrm{MA}}[1]$) and second shells ($\Delta\nu_{\mathrm{MA}}[2]$). It is assumed that these signs are consistent between samples. The fits are denoted by their combination of signs, with P for positive and N for negative. For example, a fit where $\Delta\nu_{\mathrm{MA}}[1] > 0$ and $\Delta\nu_{\mathrm{MA}}[2] < 0$ is labelled P/N, and is part of the P/N
'submodel'. Manual fits of all four submodels have been made for all three samples, at room temperature and at $420\,\mathrm{K}$. The parameters of these fits are tabulated in Supplementary section C. The N/P submodel is the only one with which qualitatively satisfactory fits could be realised for all these spectra, in the sense that the relative intensities of the subpeaks and the overall shape are qualitatively reproduced. These fits are shown in Fig. 7 (some examples of unsatisfactory fits can be found in Figs. S3 and S4). In other words, it appears that replacing an $FA^+$ cation with an $MA^+$ ion will decrease the EFG when done in the first
coordination shell, while increasing it when done in the second shell. The exact fitting parameters are also tabulated in Table 2.

Finally, the N/P fits include values for the order parameter $S$. As seen in Fig. 7, $0.2 < S < 0.35$. This is consistent with results from Grüninger et al. (2021), who measured dipolar couplings between protons of neighbouring MA and FA ions and concluded that $0.2 < S < 0.4$ for $MA_{0.25}FA_{0.75}PbI_3$ and $MA_{0.50}FA_{0.50}PbI_3$, and $0.0 < S < 0.4$ for $MA_{0.75}FA_{0.25}PbI_3$. Note





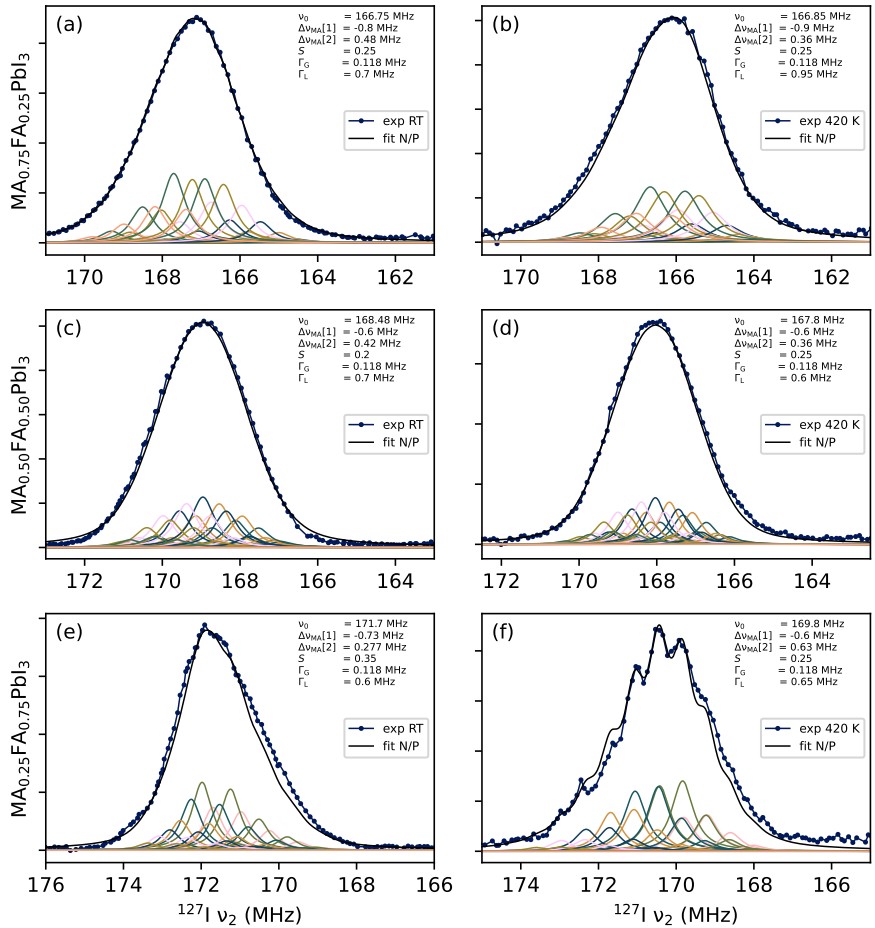

**Figure 7.** Manual fits of all mixed cation samples at room temperature (a,c,e) and 420 K (b,d,f). The shifts per first/second shell MA $\Delta\nu_{\mathrm{MA}}$ [1]/[2] are constrained to being negative/positive (N/P).

that this describes the same samples. That work, and this study, therefore both suggest there is a modest tendency for cation
species to cluster together.

It should be noted that any quantitative information, including the order parameter, should be interpreted with care. These
fits were done manually and the relation between parameters and spectral shape is fairly complex. Although an effort was
made to explore different sections of the parameter space, there is no real guarantee that there are no equivalent fits with
significantly different values. The observations regarding the number of relevant shells and the signs of $\Delta\nu_{\mathrm{MA}}[1]$ and $\Delta\nu_{\mathrm{MA}}[2]$
are more reliable, however, as they depend on qualitative features of the spectra that other versions of the model inherently
fail to reproduce. The exact values of these parameters (see again Table 2) should again be interpreted cautiously, although





**Table 2.** Manually determined optimal fitting parameters of mixed cation perovskite $^{127}$I NQR spectra at low and high temperature under the constraint that the shift per first shell MA $\Delta\nu_{\mathrm{MA}}[1]$ is negative and the shift per second shell MA $\Delta\nu_{\mathrm{MA}}[2]$ is positive. The underlying data are available at Wolffs et al. (2025).

|  | MA75 | | MA50 | | MA25 | |
|---|---|---|---|---|---|---|
| Temperature (K) | 293 | 420 | 293 | 420 | 293 | 420 |
| $\nu_0$ (MHz) | 166.75 | 166.85 | 168.48 | 168.40 | 171.70 | 169.80 |
| $\Delta\nu_{\mathrm{MA}}[1]$ (MHz) | -0.80 | -0.90 | -0.60 | -0.60 | -0.73 | -0.60 |
| $\Delta\nu_{\mathrm{MA}}[2]$ (MHz) | 0.480 | 0.360 | 0.420 | 0.360 | 0.277 | 0.630 |
| S | 0.25 | 0.25 | 0.20 | 0.25 | 0.35 | 0.25 |
| Gauss (MHz) | 0.1 | 0.1 | 0.1 | 0.1 | 0.1 | 0.1 |
| Lorentz (MHz) | 0.70 | 0.95 | 0.70 | 0.60 | 0.60 | 0.65 |

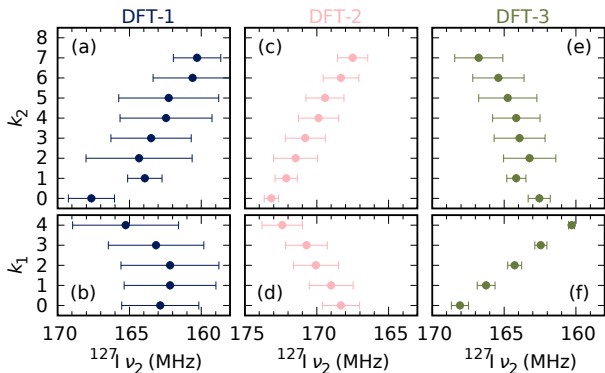

**Figure 8.** Average and root-mean-square of the simulated $^{127}$I NQR $\nu_2$ resonances in the models described in Sec. 2.2, as a function of the number of MA$^+$ ions in the first ($k_1$) and second coordination sphere ($k_2$). Note that the number of data points $n$ is quite limited: $n =$ 12, 48, 66, 60, 12 for $k_1 = 0$–4 and $n =$ 2, 2, 24, 38, 62, 32, 28, 4, 0 for $k_2 = 0$–8.

the fit for MA$_{0.25}$FA$_{0.75}$PbI$_3$ at 420 K demonstrates how the presence of highly visible subpeaks can indicate a case where $\Delta\nu_{\mathrm{MA}}[1] \approx -\Delta\nu_{\mathrm{MA}}[2]$.

### 4.5 DFT-based models

The first principles models described in Sec. 2.2 provide an independent perspective on the nature of $\Delta\nu_{\mathrm{MA}}[1]$ and $\Delta\nu_{\mathrm{MA}}[2]$. They each produce a distribution of 192 $^{127}$I frequencies that can be subdivided according to the number of first shell ($k_1$) or second shell ($k_2$) MA$^+$ ions. An overview is presented in Fig. 8, and the full result can be found in Supplementary section D. It should be noted that the statistical accuracy is limited, particularly for $k_2 = 0, 1$ and 7 and 8.





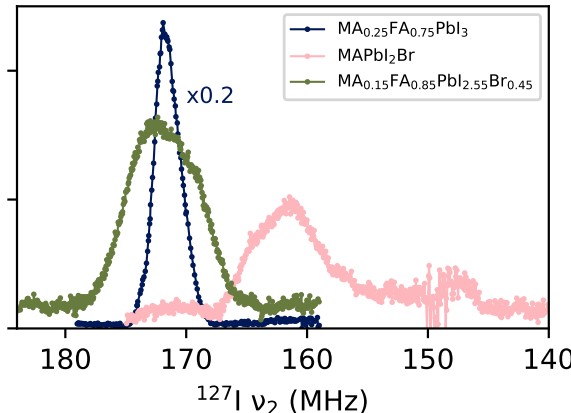

**Figure 9.** Room temperature NQR VOCS for various kinds of mixed-ion perovskites. All spectra were acquired with $2^{16}$ scans per sub-spectrum and the samples were of comparable weight ($\pm 10\%$). Below $\sim 153\,\mathrm{MHz}$ spectra exhibit distortions tentatively attributed to the transgressions of limits of the probe circuitry. The underlying data are available at Wolffs et al. (2025).

The model that is in principle most realistic, DFT-1 (Fig. 8(a-b), Fig. S5), fails to show a connection between cation coordination and frequency. The frequencies are not converged however, as not all of the relevant combinations of orientations of the twelve surrounding cations of each $^{127}$I have been sampled. Longer simulation would likely yield narrower spectra and, we hope, better resolved time-averages and clear trends, but these are computationally very expensive.

The simpler models DFT-2 and DFT-3 provide some insight at a fraction of the cost. They still produce quite broad frequency distributions, but some trends can be identified. In model DFT-2 (Fig. 8(c-d), Fig. S7) more $MA^+$ in the first (second) shell increases (decreases) the resonance frequency. In the terminology of the previous section, it has a clear P/N trend. This is correlated with the Pb-Pb distance, which is inversely related to the frequency (see also Fig. S6). This is consistent with the experimentally confirmed decrease of NQR frequencies with thermal expansion (Yamada et al., 2018). The complementary model DFT-3 (Fig. 8(e-f), Fig. S8), has the opposite trend N/P. Evidently the local effect of the cations on the anions (model DFT-3) counteracts the indirect effect from distorting of the inorganic backbone (model DFT-2). The effects are roughly comparable in magnitude, so it is not clear what the overall effect would be in more realistic models.

In summary, then, the simplified DFT models identify a phenomenon consistent with the findings of the phenomenological model but fail to confirm or reject them. To do so requires longer, expensive MD trajectories after all. It is our hope that work in this field makes these trajectories considerably more feasible. A sufficiently long MD would refine and constrain the phenomenological model, improving both its realism and ease of use.





### 4.6  $^{127}$I NQR of other ion mixes


Finally, Fig. 9 gives a preview of the type of spectra to be expected from different types of mixed perovskites. Distortions in the spectrum below the tried and tested bandwidth of 159–179 MHz exhibit spectral distortions that are tentatively attributed to the transgression of frequency-dependent limits in the transmitter/receiver electronics. It might be necessary to adjust or further increase the frequency range of the probe to study these kinds of compositions. However, it is already apparent that


a mix of halides gives rise to a significantly broader spectrum than that of mixed A site cations. After scaling intensities to compensate for the temperature, frequency and iodine content, the total integral of MAPbI$_2$Br and MA$_{0.15}$FA$_{0.85}$PbI$_{2.55}$Br$_{0.45}$ are respectively 0.8 and 0.9 times that of MA$_{0.25}$FA$_{75}$PbI$_3$, or 0.6 and 0.65 that of MAPbI$_3$ at 320 K. Given the roughness of the spectra, these numbers should be taken as overestimates. They confirm the trend from Fig. 5 that mixing ions can lead to significant signal loss. In addition, the full-width-half-maximum is more than 7 MHz in both spectra, almost three times that


of the mixed-cation samples. Still, Fig 9 establishes that NQR studies of these perovskites variants are possible, pending the elimination of low frequency distortions).

### 5  Conclusions

We have realised and demonstrated a laboratory setup making wideline NQR possible, and applied this to obtain spectra of $^{127}$I in mixed-ion lead halide perovskites. We acquired spectra of great signal-to-noise at various temperatures, enabling detailed


studies of these compositionally disordered materials. We show that the spectra of methylammonium-formadinium mixed LHPs hold a great potential for elucidating local structure and dynamics. Preliminary modelling suggests that cation substitution in the first shell around the halide has an effect on the electric field gradient on the halide opposite to cation substitution in the second shell. Simple DFT models point to two competing mechanisms, which at present prevents confirmation of the overall trend. We also identify a degree of local order consistent with previous research ($0.2 < S < 0.35$). Proving these hypotheses requires


additional research. More extensive MD calculations will be necessary to provide clearer support for the phenomenological model. In addition, the new setup allows for easy acquisition of additional compositions that will allow for the identification of clear trends.

Quickly acquired and interpretable NQR spectra should open the way towards new experiments, including but not limited to in situ measurements that are complicated to achieve in normal NMR. The combination of broadband NQR probes and


automated matching and tuning can also be very useful for other materials, not just in case of very broad spectra but also in cases where the NQR resonance is not yet known. Furthermore, it can be useful in the acquisition of NQR spectra broadened by Zeeman perturbation.

*Data availability.* The raw data and processing steps of the NQR and NMR spectra described in this publication, as well as the NQR calibration data, are stored on a repository maintained by Radboud Universiteit (Wolffs et al., 2025) and are publically available.



*Author contributions.* The laboratory setup was constructed by GJ and calibrated by JW. NQR measurements were performed by JW with support from JG. NMR measurements were performed and analysed by JW and JG. DFT calculations were performed and analysed by GW. JW, JG and AK were the principal in planning and organising the project. The manuscript was principally written by JW with input from all other authors, in particular from GW with regards to the DFT calculations. AK conceived and supervised the project.

*Competing interests.* One of the authors (AK) is a member of the editorial board of journal Magnetic Resonance. The authors declare no
other competing interests.

*Acknowledgements.* We thank the group of dr. Helen Grüninger at Universität Bayreuth for providing the samples, and dr. ir. Menno Bokdam for the use of his molecular dynamics trajectory. The facilities technicians ing. Hans Janssen and ing. Ruud Aspers are thanked for their support.

*Financial support.* T

he Dutch Science Council (NWO) is acknowledged for the support of the solid-state NMR facility for advanced materials science, which is part of the uNMR-NL ROADMAP facilities (NWO project no. 184.035.002). The Dutch Science Council (NWO) is acknowledged for the support of the solid-state NMR facility for advanced materials science, which is part of the uNMR-NL ROADMAP facilities (NWO project no. 184.035.002).





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
