# Peer review of "Automated wideline nuclear quadrupole resonance of mixed-cation lead halide perovskites"

_Magnetic Resonance, 2025_

## Author Response (AR1)

MR-2025-2 Response to reviewers

**Anonymous Referee 1**

We thank Anonymous Referee 1 for the positive evaluation of our work and the detailed comments for improvement. Our point-by-point answers to the comments are:

*Formula (1) of page 2 is correct and provides the NQR frequencies for a spin I = 5/2 nucleus when h = 0, as stated by the authors. There is nothing inherently incorrect in the statements made by the authors. However, I worry that this may confuse readers new to the field of NQR, as eta (h) will influence the transition frequencies when η ≠ 0. I suggest adding a statement to clarify that the NQR frequencies are influenced when h ≠ 0.*

We agree with the reviewer and added a statement to clarify that the NQR frequencies are influenced when η ≠ 0 on page 2 of the revised manuscript.

*On page 2, line 35: The authors state that NQR has re-emerged partly due to lead halide perovskites. I suggest to the authors to add a short line explaining what NQR spectroscopy offers from other techniques, such as solid-state NMR and X-ray diffraction.*

At the end of page 2 we have added a few sentences about the specific advantage of halide NQR to study LHPs compared to NMR and XRD.

*On page 3, line 55: The authors state that the NQR spectrum becomes "too wide to easily acquire with regular, commercially available equipment". To my knowledge, there are no regular commercial NQR instruments that are available, and commercial solid-state NMR console (Bruker, Varian, Tecmag) can be adapted to NQR. I suggest rephrasing this to simply saying that the broader NQR spectra are even more difficult to observe. The "commercial" aspect may not be appropriate here, as current solid-state NMR consoles are sufficiently flexible to be adapted to a specific challenge, such as the case of this work where a Varian NMR console was used.*

We agree with the reviewer and removed the reference to commercial equipment, now only pointing out the technical challenge to record extremely wideline NQR spectra of LHPs with occupational disorder.

*On page 6, line 163: The authors cite Pyykko's paper from 2008, but there has since been a 2017 update. The differences in the quadrupole moment of 127I is not large so it is not worth repeating the calculations for this small correction, but perhaps worth noting for future work. Unless there was a reason why the 2008 value was chosen.*

We agree with the reviewer that future calculations should use the newer quadrupole moment, but the reported difference in the quadrupole moment of 127I is only 1.2%. We have therefore not repeated the calculations and left the article unmodified on this point.

*On page 6, line 166: Could the authors please define the number of turns, pitch (distance between turns), and wire gauge used to construct the NQR coil? I also suggest reporting the probe's Q or the bandwidth of the resonator (this is distinct from the bandwidth of the probe), but I will leave the latter up to the author's discretion.*

We have added a detailed description of the coil to the description of the wideline NQR setup on page 6, line 171 we now give the probe Q, and the number of turns, length and wire gauge.

*The authors measured the v2 NQR frequency of 127I. There is a clear benefit to this approach in terms of s/n. I suggest clarifying in the manuscript why v2 was chosen rather than v1.*

Indeed we clearly chose v2 for the better S/N at the higher frequency and clarified our choice for the second NQR resonance in section1.3, line 65.

*Page 18, line 312: The authors mentioned distortions twice in the sentence: "Distortions in the spectrum below the tried and tested bandwidth of 159–179MHz exhibit spectral distortions that are tentatively attributed …" which could be clarified.*

We thank the referee for spotting the error and have rephrased the sentence (page19 line 336).

*As broad NQR lines in mixed halide perovskites have been acquired using similarly automated NQR instrumentation (Aebli et al., 2021), to my understanding, the most novel part of this work appears to be in the model used to interpret the NQR data. Could the authors clarify the novel aspect for the hardware in the manuscript?*

With apologies to both the referee and the authors of Aebli et al. (2021) if we are wrong, but it is our impression that the VOCS experiments in the Aebli paper were done without automation. There is no reference to the use of an ATM system or any means of automation in the paper. However, we acknowledge that the experiments by Mozur et al (2020) are using an ATM which we explicititly reference in line 61. Unfortunately Mozur et al. did not provide details about the approach to record undistored spectra in an automated way. We have added a detailed description of our automated approach on page 10 of the revised manuscript; we added VT operation, lineairized the rf-field strength, and adapted the order in which record the spectra at different frequencies to get a reliable performance from the ATM thus recording overall lineshapes that are not distorted by technical imperfections. Finally, we have separated the operation of the ATM from the pulse program so that remote operation and corrections thereof are facilitated. Therefore we believe to have improved upon the NQR approach in a way that contributes to the field.

**Anonymous Referee #2**

We thank anonymous reviewer #2 for the positive evaluation of our work and the suggestion for improvement:

*In the experimental section on NQR (page 8), it would be helpful to include the dimensions of the sample holder and specify whether the sample was positioned at the center of the RF coil to minimize RF inhomogeneity.*

We have added details about the coil and the requested details on sample holder dimensions and sample placement on page 8 (~line 200) of the revised manuscript.